# Prevalence of Ponticulus Posticus and Migraine in 220 Orthodontic Patients: A Cross-Sectional Study

**DOI:** 10.3390/biology12030471

**Published:** 2023-03-20

**Authors:** Monica Macrì, Fabiola Rendina, Beatrice Feragalli, Francesco Pegreffi, Felice Festa

**Affiliations:** 1Department of Innovative Technologies in Medicine & Dentistry, University “G. D’Annunzio” of Chieti-Pescara, 66100 Chieti, Italy; 2Department for Life Quality Studies, University of Bologna, 40126 Bologna, Italy; 3Department of Biomolecular Sciences, University of Urbino Carlo Bo, 61029 Urbino, Italy

**Keywords:** ponticulus posticus, pain, CBCT, arcuate foramen, atlas, migraine, Dolphin software, orthodontics

## Abstract

**Simple Summary:**

The vertebral artery passes through the first cervical vertebra (or atlas). Around 20% of patients carry an atlas mutation in the presence of ponticulus posticus, an additional bony ridge over the vertebral artery groove that compresses or irritates the artery with its sympathetic periarterial plexus and suboccipital nerve. Early atlas mutation identification is essential for accurate diagnosis in patients affected by unexplainable symptoms, such as headache, neck pain, vertigo, visual disturbances, problems with speech and swallowing, neurosensory hearing loss, vertebrobasilar insufficiency, dizziness, postural muscle tone loss, shoulder pain, etc. Our cross-sectional study aimed to assess the prevalence and morphologic features of ponticulus posticus and its association with migraine using cone beam CT scans. We confirmed in a homogeneous population (220 Italian adults) the association between migraine and ponticulus posticus.

**Abstract:**

Background: Ponticulus posticus (PP) is a medical term that describes an anomaly of the atlas (C1), which has a complete or partial bone bridge over the vertebral artery (VA) groove. The purpose of the study is to estimate the prevalence of PP in patients with a diagnosis of migraine. Methods: Cone beam CT (CBCT) scans (n = 220) were reviewed for the detection of PP in the University “G. D’Annunzio” of Chieti in the Department of Medical, Oral and Biotechnological Sciences. The sample included 220 Italian patients between 18 and 87 years. Pearson chi-square analysis (*p* < 0.05 and 95% CI) was used to establish an association between migraine and PP. Results: The present study found a prevalence of PP of 20.9% and a prevalence of migraine of 12.272%. The association between migraine and PP was confirmed by the chi-square statistic, since the *p*-value was 0.008065 (significant at *p* < 0.05). PP was more frequent in the migraine without aura group, without a statistical difference relative to the migraine with aura group. Conclusions. The study concluded that PP is positively associated with migraine.

## 1. Introduction

Ponticulus posticus (PP) is a medical term adopted from Latin, meaning “little posterior bridge”, also known in the literature by other terms: Kimmerle’s anomaly, pons ponticus, canalis arteriae vertebralis, posterior atlantoid foramen, foramen sagittal, atlas bridging, arcuate foramen, posterior ponticulus, foramen retroarticular and retroarticular ring [1].

In humans, the cervical vertebrae are the vertebrae of the neck (from C1 to C7) positioned between the skull and the subsequent thoracic vertebrae. The cervical vertebrae are seven vertebrae characterized by their small size. C1 is the first vertebra immediately below the skull, also named vertebra cervicalis I or the atlas bone. Atlas is a medical term derived from a Titan in Greek mythology called Atlas, who supported the earth on his shoulders for all eternity as a punishment by Zeus. Instead, the first cervical vertebra supports the skull’s weight, also called the atlas bone, as described in Greek mythology. The atlas is a bony ring consisting of an anterior arch, a posterior arch and two lateral masses, and it lacks a vertebral body and spinous process. Each of these two lateral masses has an upper and a lower articular surface, which articulates, respectively, with the occipital condyles and the second cervical vertebra, commonly called the axis (C2).

The primary arteries of the neck, the vertebral arteries, branch off from the subclavian arteries. Each vertebral artery enters the transverse process at the level of C6 and ascends into the transverse foramen of each cervical vertebra. It then leaves the transverse foramen of C1 and travels through the posterior arch of C1 before entering the foramen magnum. The basilar artery is formed inside the skull when the right and left vertebral arteries combine and supply the cerebellum, the brainstem and the posterior brain regions.

PP is a congenital anomaly of the atlas (C1) characterized by a complete or incomplete bony ridge over the vertebral artery groove (VAG). It occurs due to calcification of the oblique atlantooccipital ligaments (OAOLs, arcuate ligaments). The vertebral artery (VA) [2], sympathetic periarterial plexus and suboccipital nerve are situated in this area and may compress or be irritated by the presence of PP.

The clinical significance of PP is controversial because several studies have reported symptomatologic conditions associated with PP. Nonetheless, the majority of patients are asymptomatic [3].

The painful symptoms may be related to vertebral artery compression, suboccipital nerve compression [4] or sympathetic periarterial plexus irritation, as supposed in the literature.

Detection of this anomaly is essential as PP may be involved in the development of otherwise unexplainable symptoms widely reported in the literature: headache, neck pain, vertigo, swallowing and speech problems, visual disturbances, neurosensory hearing loss, vertebrobasilar insufficiency, dizziness, postural muscle tone loss, shoulder pain, Barré–Liéou syndrome, vascular issues and loss of consciousness [5].

Numerous studies have detected PP by CT and lateral cephalography and described cone beam CT (CBCT [6]) as the gold standard technique to identify the presence of PP [7,8,9]. 

Unpredictable episodic migraine attacks usually occur on one side of the head. Autonomic symptoms such as nausea, photophobia and phonophobia can occur with migraine [10]. Less than one third of the cases suffer a perceptual disturbance called “aura” before the head pain and other symptoms. The International Classification of Headache Disorders (ICHD) has classified two types of migraine [11]:○with aura;○without aura (formerly called classic and common migraine).

The International Headache Society (IHS) defines migraine as a recurring headache on one side of the head that can be moderately or extremely painful, throbbing or pulsating. A single migraine episode lasts 4 to 72 h, or less in the pediatric population (1 to 72 h). Migraine is a medical condition that might last a lifetime.

The etiology of migraine has been extensively studied, but the exact causes are still unknown; genetics play a significant role in migraine.

Migraine is triggered by the stimulation of a neurological pathway that induces the release of pain-producing inflammatory substances. These chemical signals temporarily affect the cerebral nerves and vessels due to abnormal brain activity.

A migraine diagnosis is mainly clinically based on medical history, symptoms and a physical and neurological examination by a physician specialized in treating headaches (i.e., neurologist). However, migraine is misdiagnosed and undertreated in most cases.

This paper, therefore, aims to investigate the prevalence and morphologic features of PP and its association with migraine in 220 patients using CBCT scans.

## 2. Materials and Methods

### 2.1. Study DesignSS

The present work is a cross-sectional study to retrospectively determine the presence of PP on CBCT scans and the possible association with migraine in 220 adult patients.

### 2.2. Setting (Setting, Locations and Relevant Dates, including Periods of Recruitment, Exposure, Follow-Up and Data Collection)

The clinical data included in this cross-sectional study come from the Department of Medical, Oral and Biotechnological Sciences archives in the University of Chieti “G. D’Annunzio”, selected within a time range from 2018 to 2022. The studies adhered to the European Union Good Practice Rules according to the Helsinki Declaration. An Independent Ethics Committee approved the protocol with number 23 in the Hospital of Chieti.

### 2.3. Participants

This cross-sectional study enrolled patients between 18 and 87 years old, treated in the Department of Orthodontics and Orofacial Pain for malocclusions or temporomandibular disorders (TMD) [12]. 

The cohort included 220 Italian patients aged 18 to 87 years, and the overall mean age was 47.6 ± 26.5 years, composed of 110 male patients (50%) and 110 female patients (50%), to ensure a homogeneous sample in terms of sex and gender. The age of male patients ranged from 18 to 86 years (mean ages ± standard deviation; 50.1 years ± 17.78), and it ranged from 18 to 87 years (41.6 years ± 20.56) in females.

All patients in the study provided written informed consent and presented medical information with details about migraine (i.e., presence of absence of aura, episodic or chronic attacks) as preliminary steps before the specific cranial/cervical musculoskeletal clinical examination and the radiological examination, consisting of CBCT. 

### 2.4. Exclusion Criteria

The exclusion criteria were as follows: ○Age under 18;○Non-Italian nationality;○Low-quality images (e.g., due to patient movement);○Incomplete region of interest (ROI);○Head congenital anomalies or syndromic pathologies;○Anamnesis positive for trauma or surgery of the neck;○Anamnesis positive for traumatic injury or concussion of the brain;○Anamnesis positive for more than 1 type of primary headache; ○Anamnesis positive for comorbidity with other medical conditions (e.g., epilepsy, diabetes mellitus, underlying hypertension, cerebrovascular diseases or other neurodegenerative disorders);○Anamnesis positive for psychiatric conditions (anxiety disorders, schizophrenia bipolar disorders or major depressive disorders) [13];○Inability to provide a detailed history. 

### 2.5. Diagnosis of Migraine

Only patients with a diagnosis of migraine performed by a doctor specialized in treating headaches (i.e., neurologist) after a neurological examination were enrolled in the current study. Migraine diagnosis is mainly based on the symptoms described by the patient due to the absence of any specific diagnostic tests. In addition, medical history, physical examination, fulfilment of the diagnostic criteria and exclusion of any other causes of the pain in the head region are relevant for migraine diagnosis [14].

### 2.6. CBCT Scans

All CBCT scans were acquired with a low-dose protocol (FOV of 240 × 190, acquisition time of 15 s, 80 kVp, 5 mA, 35 μSv) [15] using the Planmeca Promax^®^ 3D MID unit (Planmeca Oy, Helsinki, Finland).

During the CBCT exams, the head of the patient was oriented in a natural head position (NHP). NHP is a morphological analysis used in orthodontics and described in the anthropological literature as a reproducible and physiological head posture in a tridimensional space [16,17]. To obtain the NHP, all the patients looked into their own eyes through a frontal mirror positioned at a distance of 1.5 m. The patients were sitting with the back perpendicular to the floor, and the head was stabilized with ear rods in the external auditory meatus [16,17].

After acquiring the CBCT data in the Dolphin software, the head was additionally oriented according to the three reference planes on the front, right and left views [16,17]. The cephalometric measurements were correctly performed using the widgets present in the software [16,17].

### 2.7. Evaluation of PP

In the posterior arch of the atlantal arch (C1), the vertebral artery produces an impression on the superior surface, and this groove is called the vertebral artery groove (VAG) [18]. The presence of PP is an anomaly that is radiographically detected with a complete bone bridge over the VAG (Figure 1, Figure 2 and Figure 3). However, the partial form of PP is more common than the complete form and is radiographically detected as a bony spike above the VAG (Figure 4). This cervical anomaly is diagnosed radiographically, and the literature has reported CBCT as the gold standard tool for diagnosing PP. The prevalence study of PP is influenced by the quality and resolution of the radiographic examination adopted, and the cervical anomaly can be complete or partial, unilateral or bilateral. Therefore, three-dimensional images such as the CBCT exam are more accurate than two-dimensional examinations such as latero-lateral teleradiography of the head. The accuracy of the CBCT scans allowed us to exclude radiographic findings similar to the appearance of PP, as in the case of deep and accentuated bone angles of the vertebral artery groves. CBCT evaluation, therefore, makes the prevalence study of PP more accurate.

### 2.8. Error Method

The reliability of this research was increased by randomly analyzing and selecting all the CBCT images. At most, 50 scans/die were checked to reduce the risk of inaccurate analysis due to fatigue. Two different operators analyzed all the data twice; in case of disagreement, a third observer was consulted to confirm the type or absence of PP. This method improved the repeatability and reproducibility of the analysis and reduced intraoperator and interoperator errors. The reliability between operators 1 and 2 was determined with Cohen’s* Kappa, which showed strong significant agreement (k = 0.87).

### 2.9. Statistical Analysis

The prevalence study of PP and the possible association with migraine were performed with statistical analysis based on individual patient data entered in a spreadsheet (Microsoft Excel version 16.0 for Windows 10). The collected data were analyzed using statistical analysis software (Stata 9.0 software of StataCorp LP, College Station, TX, USA, IBM SPSS 17 software of IBM Corporation, New York, NY, USA). The data analysis was compared with the data reported in Table 1 using Pearson’s chi-squared test (*p* < 0.05 and 95% CI).

## 3. Results

CBCT scan data for 220 patients (110 females and 110 males) from 18 to 87 years old were included in the study. 

As reported in Table 1, 46 patients (20.9%) had a diagnosis of PP, and 27 patients (12.272%) were positive for migraine according to the exclusion criteria. Analyzing the prevalence of migraine in the female group, we found 17 cases out of 110 (15.154%), and there were 10 males (9.090%), with an F/M ratio of 1.7. The prevalence of migraine was additionally investigated in the group positive for PP, as reported in Table 1. 

The PP cases were additionally analyzed according to the classification, as reported in Table 2.

The chi-square statistic confirmed the association between migraine and PP since the result was 43.2795 and the *p*-value was <0.00001 (significant at *p* < 0.05). PP was more frequent in patients with migraine without aura (32.608%) than in patients with migraine with aura (19.565%). No statistical difference between migraine with and without aura in different categories of PP (*p* > 0.05) was found, as reported in Table 2.

## 4. Discussions

This paper evaluated the associations between PP and migraine in adult patients (18–87 years) treated in the Department of Orthodontics and Orofacial Pain for malocclusions or TMD who underwent a CBCT exam during the first visit.

Early atlas mutation identification is important for a comprehensive diagnosis in patients affected by recurring headaches and other unexplainable neurosensory symptoms in the shoulder, neck and head. The first objective of this work was to assess the prevalence of PP using cone beam CT scans; the second objective was to determine the association with migraine by analyzing the data from the patients’ medical records.

We retrospectively analyzed the posterior arch of the first cervical vertebra by CBCT as the gold standard tool for diagnosing PP, resulting in an accurate study of the prevalence of this condition. We selected 110 male and 110 female patients to ensure a homogeneous cohort in terms of gender and sex. The CBCT exam was performed using the low-dose protocol to decrease the patient’s radiation exposure. PP is part of various congenital atlantal arch malformations, and knowledge about it and its radiological appearance is important to identify other correlated symptomatologic conditions. Overall, PP is considered as benign anatomical variations detected incidentally by radiographic exams. PP is visible on the posterolateral margin of the atlas as a complete or incomplete bony ridge over the vertebral artery groove due to calcification of the oblique atlantooccipital ligaments. However, rare sensory or neurological symptoms or deficits have been associated with this anomaly. PP and most congenital anomalies of C1 are found incidentally in asymptomatic patients.

Migraine is usually classified with or without aura. In the literature, many headache clinicians and researchers have investigated these two types of migraine as different manifestations of the exact underlying pathophysiology or as different entities, so the anatomic origin of migraine was also investigated with advanced imaging techniques, such as fMRI.

Migraine pain could originate from an episode of local sterile meningeal inflammation that causes the activation of trigeminal primary afferent nociceptive neurons [19]. As reported in the literature, neck muscle inflammation promotes the sensitization of primary trigeminal neurons and is considered a risk factor associated with migraine pathology [20]. The purpose of many studies cited in this paper was to investigate whether the presence of PP may trigger many symptoms of vertebrobasilar insufficiency (i.e., headache, migraine) caused by the irritation of the vertebral artery.

As reported in the literature, PP is detected in quadrupeds or early primates as an additional lateral extension for attachment of the posterior atlantooccipital membrane [21].

Adopting an upright posture in bipeds led to the progressive disappearance of the PP, probably because the neck muscles and ligaments support the head, and the vertical load is transferred from the head to the upper articular atlas process [22].

The copresence of PP and migraine was found in 12.272% of the recruited patients, and the statistical analysis confirmed that migraine and PP are significantly associated since *p* < 0.05; as reported in Table 1, there was a highly significant difference in the prevalence of PP for patients with migraine compared with patients without migraine. As previously described by Wight [23], chiropractic manipulation of the atlas results in a considerable improvement in the treatment of migraine, and there was a significant association with PP in patients with migraine without aura but not in patients with migraine with aura.

No difference was found in the prevalence of PP for patients with migraine and aura compared to those without migraine. Similar results were described in the literature by Sabir et al. [24], who found a positive association between PP and migraine, concluding that this result is related to the occlusion of the vertebral artery caused by PP.

Mokhtari et al. [25]. found migraine in only 3.4% of patients without PP anomaly and a significant association between PP and migraine.

Other studies found more severe neck pain and vertigo in patients with complete PP (Cakmak et al. [26] and Ratnaparkhi et al. [5]).

In this study, the prevalence of PP was 20.9 %. This is comparable to the results of similar published articles, which ranged between 5.1% and 37.8% in the Western population [27].

Similar to the findings of the published literature, bilateral PP (12.7%) and complete PP (10.3%) were found to be rare.

Since the prevalence of PP is not correlated with gender or sex, as found in the literature, the present paper did not investigate this aspect [28].

As reported by Burch [29], migraine affects an estimated 12% of the population. According to our findings, the prevalence of migraine was 12.272% in this study. In the Caucasian population, migraine is more frequent than in the Asian population [30]. Due to this racial difference, this study only recruited Italian people in the sample. The majority of studies focused on migraine evidenced the existence of gender differences; the prevalence is approximately twice as high in females compared with males [31,32,33]. Consistent with the literature, the prevalence of patients positive for migraine was higher in females than males, with an F/M ratio of 1.7.

The present study only enrolled adult patients at least 18 years of age, since the PP prevalence in terms of age is controversial due to the etiological hypothesis of progressive calcification of the atlantooccipital ligament with progressive mineralization over time [34]. Few papers in the literature have suggested the PP outcome of senile ossification [35]; however, the association between PP and chronological age was rejected in many studies [27]. The main limitation of this study is represented by the broad age range of the sample, since it was not homogeneous, ranging from 18 to 87 years. CBCT evaluation of PP provides a novel, valuable and reliable predictor for the diagnosis of migraine. Associated with the wide age range, a second limitation is represented by the causes of migraine, which have not been investigated in this work due to the broad variety of possible migraine triggers as suggested in the literature, including hormonal, emotional, physical, dietary, environmental and medicinal factors.

The skull, meninges, teeth, neck and shoulder have the same embryological origin, and their development is influenced by neural crest cells (NCCs), as described in the literature. As NCCs are a multipotent embryonic cell population that contributes to the formation of the sella turcica, teeth and atlas, if a mutation occurred in this stage, two or more congenital anomalies could be present [36]. Some dentoskeletal anomalies in the midface and neck area, such as maxillary canine impaction (MCI), atlas posterior arch deficiency (APAD) and sella turcica bridging (SB), and lateral incisor anomalies and their possible correlation with PP were analyzed by some authors due to the same or similar genetic origin [27,37,38]. The sella turcica region is an important landmark during cephalometric analysis, so it is easy to detect by orthodontists [39,40,41].

There may be other cervical anomalies in the atlas that deserve further study. In this work, we retrospectively reviewed the cervical CBCT of 220 patients. We identified other congenital anomalies of C1, such as occipitocervical synostosis and congenital anomalies of the posterior arch of the atlas [39,40,41]. Our future goal will be to determine the prevalence of these other anatomic C1 variations and their possible association with migraine [39,40,41]. Occipitocervical synostosis is a congenital fusion of C1 with the base of the occipital bone. It may be clinically relevant, as it may compress the brainstem, vertebral artery and cranial nerves due to the narrowing of the foramen magnum [39,40,41,42]. This variant is also known in the literature as occipitalization of the atlas, atlantooccipital fusion or atlantooccipital assimilation [43]. A congenital defect of the posterior arch of the atlas is classified into five severity forms ranging from aplasia (absence of the entire arch including the tubercle) to simple clefts because of the incomplete posterior fusion of the two hemiarches above the midline. Dentists performing CBCT should be familiar with and identify these cervical abnormalities and avoid misdiagnosing them as fractures [44,45]. These anatomical variants of the atlas may cause pain in the cervical and head regions, cerebrovascular accidents and more. Despite the numerous benefits of cone beam computed tomography (CBCT), its applications to the medical field are limited, mainly due to poor image quality in the past. Enhancing the CBCT image quality has a crucial impact on diagnostic accuracy and patient safety in the study of head and neck anatomy.

Exhaustive knowledge about these anomalies in the cervical area is required of neurosurgeons, maxillo-facial surgeons and otolaryngologists to avoid misdiagnosis and any complications during surgery [46], especially during the placement of lateral mass screws into the atlas.

## 5. Conclusions

This cross-sectional study found a positive association between PP and migraine.

Dentists can consider migraine a risk factor for PP, and the radio-detection of PP is required because this anomaly may be associated with other symptoms.

In light of the prevalence of PP and of the positive associations with migraine, when a migraine remains without a correct diagnosis and clinical management, cooperation between various medical specialists (i.e., headache specialists, neurologists, oral doctors, pharmacologists, rehabilitation specialists, internal medicine doctors) may be necessary, and we may consider the CBCT evaluation of PP as a novel and valuable predictor for the diagnosis of migraine. According to the results, PP detection would reduce migraine misdiagnosis and undertreatment.

## Figures and Tables

**Figure 1 biology-12-00471-f001:**
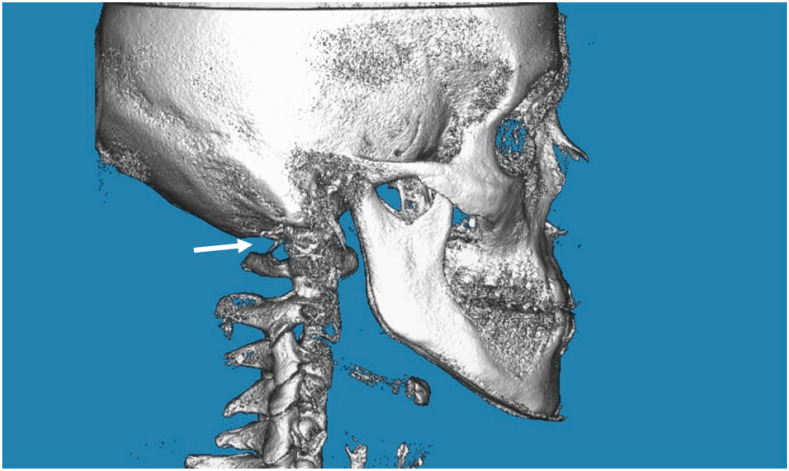
CBCT image of the complete PP on the right (indicated by the arrow). The image is taken from the “G. D’Annunzio” University archives.

**Figure 2 biology-12-00471-f002:**
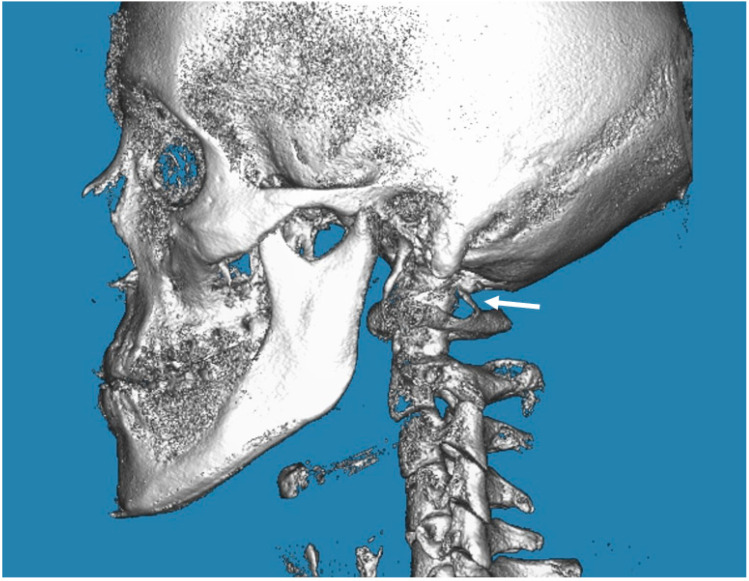
CBCT image of the complete PP on the left (indicated by the arrow). The image is taken from the “G. D’Annunzio” University archives.

**Figure 3 biology-12-00471-f003:**
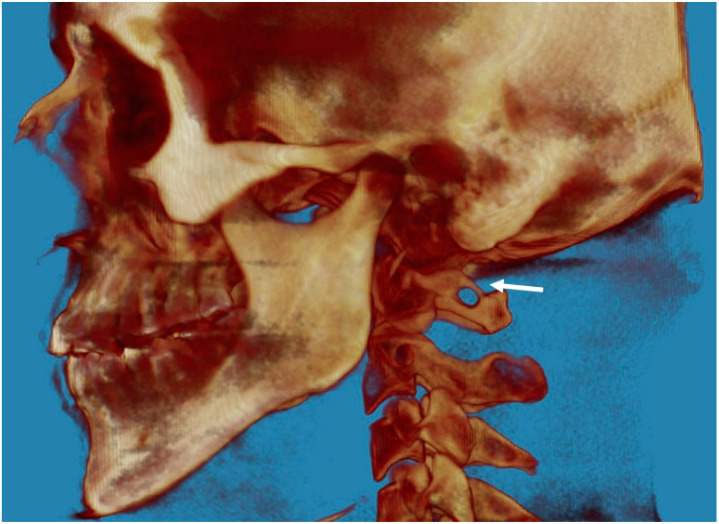
CBCT image of the complete PP on the left (indicated by the arrow). The image is taken from the “G. D’Annunzio” University archives.

**Figure 4 biology-12-00471-f004:**
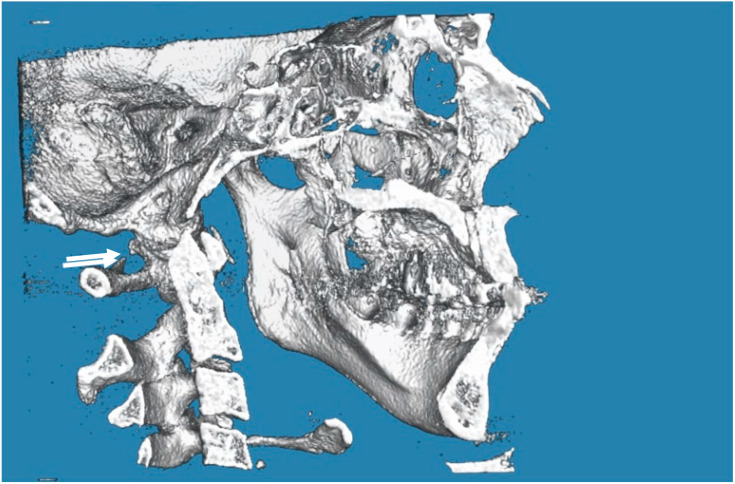
CBCT image of the partial PP on the left (indicated by the arrow). The image is taken from the “G. D’Annunzio” University archives.

**Table 1 biology-12-00471-t001:** Evaluation of PP and migraine in participants.

	% of Population	% of Group with PP
Migraine without aura	5.000% (N 11)	32.608% (N 15)
Migraine with aura	7.272% (N 16)	19.565% (N 9)
Migraine absent	87.272% (N 193)	47.822% (N 22)
Marginal column totals	(N 220)	(N 46)

**Table 2 biology-12-00471-t002:** Evaluation of PP and migraine in participants.

	Migraine with Aura	Migraine without Aura	Migraine Absent	Marginal Row Totals
Bilateral complete PP	10.869% (N 5)	8.695% (N 4)	8.695% (N 4)	28.260% (N 13)
Complete PP on the left	2.173% (N 1)	0.000% (N 0)	4.347% (N 2)	6.521% (N 3)
Complete PP on the right	0.000% (N 0)	0.000% (N 0)	4.347% (N 2)	4.347% (N 2)
Complete PP on the right and partial on the left	2.173% (N 1)	2.173% (N 1)	2.173% (N 1)	2.173% (N 3)
Complete PP on the left and partial on the right	0.000% (N 0)	0.000% (N 0)	4.347% (N 2)	4.347% (N 2)
Partial bilateral PP	10.869% (N 5)	6.521% (N 3)	15.211% (N 7)	32.595% (N 15)
Partial left PP	4.347% (N 2)	2.173% (N 1)	4.347% (N 2)	10.869% (N 5)
Partial right PP	2.173% (N 1)	0.000% (N 0)	4.347% (N 2)	6.521% (N 3)
Marginal column totals	32.595% (N 15)	19.557% (N 9)	47.806% (N 22)	(N 46)

## Data Availability

Not applicable.

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
