# Peer review of "Prevalence of Ponticulus Posticus and Migraine in 220 Orthodontic Patients: A Cross-Sectional Study"

_biology, 2023, doi:10.3390/biology12030471_

Round 1

Reviewer 2 Report

This study identified the correlation between the presence of PP and predisposition to migraine. Even though PP was extensively investigated by numerous studies, its association with migraine was not. There are a few comments to improve the paper. 

Why was the sample size of 220 used? Please show how appropriate sample size was calculated?

Was there any association between the presence of PP and sex or age? Sex difference seems statistically significant to me.  

There are other variants and defects that could have been studied such as atlas arch defects. Were these defects studied? If not, it could be the authors’ next study.

There are several synonyms used to describe PP and the foramen formed by the arch such as foramen arcuale, Kimmerle’s anomaly, and etc. Please add these to the discussion and/or introduction.  

How is the PP formed? Please add to the discussion on how the PP is formed.  

In the conclusion section, please avoid the word “radiograph” because CBCT was used in this study.

For all figures, please add arrows or arrow heads to point the PP. For Figure 4, it seems that the arch is just the boundary of the transverse foramen. Please verify. 

References and citations are not properly formatted according to MDPI guidelines.

Round 2

Reviewer 1 Report

In my opinion the manuscript in this form is publishable